# A Uni-Micelle Approach for the Controlled Synthesis of Monodisperse Gold Nanocrystals

**DOI:** 10.3390/nano14110900

**Published:** 2024-05-21

**Authors:** Liangang Shan, Wenchao Wang, Lei Qian, Jianguo Tang, Jixian Liu

**Affiliations:** Institute of Hybrid Materials, National Center of International Research for Hybrid Materials Technology, College of Materials Science and Engineering, Qingdao University, Qingdao 266071, China; shanliangangqdu@163.com (L.S.); wangwenchao1021@163.com (W.W.); q_l666@163.com (L.Q.)

**Keywords:** small-size alloy nanoparticles, monodisperse gold nanocrystals, uni-micelle, reversible phase transfer

## Abstract

Small-size gold nanoparticles (AuNPs) are showing large potential in various fields, such as photothermal conversion, sensing, and medicine. However, current synthesis methods generally yield lower, resulting in a high cost. Here, we report a novel uni-micelle method for the controlled synthesis of monodisperse gold nanocrystals, in which there is only one kind micelle containing aqueous solution of reductant while the dual soluble Au (III) precursor is dissolved in oil phase. Our synthesis includes the reversible phase transfer of Au (III) and “uni-micelle” synthesis, employing a Au (III)-OA complex as an oil-soluble precursor. Size-controlled monodisperse AuNPs with a size of 4–11 nm are synthesized by tuning the size of the micelles, in which oleylamine (OA) is adsorbed on the shell of micelles and enhances the rigidity of the micelles, depressing micellar coalescence. Monodisperse AuNPs can be obtained through a one-time separation process with a higher yield of 61%. This method also offers a promising way for the controlled synthesis of small-size alloy nanoparticles and semiconductor heterojunction quantum dots.

## 1. Introduction

The gold nanoparticles (AuNPs) have received widespread attention in various fields, including photothermal conversion [1], medical detection [2,3,4], catalysis [5,6,7,8,9], and sensing [10,11], which are attributed to the small-size effect and localized surface plasmon resonance (LSPR) effect [12,13]. There are several methods for synthesizing AuNPs [14,15], including a phase transfer [14], microemulsion [16], biosynthesis [17,18], etc. However, some of these methods involve the use of toxic agents, such as toluene and hexadecyltrimethylammonium bromide, which can result in chemical contamination [19,20]. McDonagh et al. [21] have synthesized highly thermally stable AuNPs using a phase transfer, but the separation of the products by rotary evaporation is necessary, which leads to difficulties in the removal of oleylamine (OA) [22]. In addition, the extensive use of OA for the reduction of AuNPs tends to increase costs. Pileni et al. [23] first reported the conventional microemulsion method in the 1980s; we call it a bi-micelle process, which was carried out by mixing two types of micelles containing different precursors separately through an intermicellar exchange (Appendix A). The nanoparticles synthesized with the bi-micelle process are polydisperse due to the micellar coalescence, which needs a multiple size-selective extraction/precipitation process [24] (more than three times the separation) for reducing the size distribution, resulting in a lower yield and high cost.

Therefore, it is crucial to develop a new method for synthesizing AuNPs with precise size control and a lower cost. Here, we report a novel uni-micelle method for synthesizing size-controlled AuNPs. First, Au (III) is endowed with dual solubility in water/oil by combining it with amine, and Au (III) can diffuse between water/oil, which is a reversible phase-transfer process. Then, “uni-micelle” synthesis was constructed, employing gold–amine complexes (Au (III)-OA) as an oil-soluble precursor, and water-soluble ascorbic acid (VC) is chosen as a reducing agent, sodium bis(2-ethylhexyl) sulphosuccinate (AOT) as surfactant, and isooctane as the oil phase, constructing the uni-micelle system. The size-controllable synthesis of AuNPs can be achieved by tuning the ratio of water to AOT (*w*) due to the depressed micellar coalescence [21]. The method will provide a general protocol for synthesis of small-size alloy nanoparticles and semiconductor heterojunction quantum dots.

## 2. Materials and Methods

### 2.1. Materials

Dodecylthiol (98%), oleylamine (OA, 80–90%), hexylamine (99%), n-Octylamine (GC), decylamine (98%), dodecylamine (98%), isooctane (AR), glucose (98%), ascorbic acid (AR), and ethanol were purchased from Macklin. Silver tri-fluoroacetate (98%), hydrogen tetrachloroaurate(III) (HAuCl_4_·4H_2_O, 99.0%), sodium sulfide (Na_2_S, 99.9%), and sodium bis(2-ethylhexyl) sulfosuccinate (AOT, 98.0%) were purchased from Aladdin Biotechnology Technology Co., LTD (Shanghai, China) Water was purified with a Millipore water system (ZIQ7000T0C, Merck Co., LTD, Beijing, China).

### 2.2. Preparation of Precursor Solution

The oil phase solution of the precursor (Au(III)-OA) was prepared using a reversible phase-transfer method at ambient temperature and pressure. An amount of 0.8240 g of chloroauric acid tetrahydrate was dissolved in 20 g of distilled water to prepare a 0.10 mol/L chloroauric acid aqueous solution, which was then diluted from 1 mL to 10 mL and added to a 10 mL solution of oleylamine ethanol with a concentration of 0.05 mol/L, followed by stirring for half an hour. After 10 mL of isooctane was added into the mixture and stirred for half an hour, the supernatant was centrifuged to obtain the isooctane solution of Au(III)-OA. To prepare a 0.60 mol/L ascorbic acid solution, dissolve 0.3170 g of ascorbic acid in 3 g of distilled water. Dissolve 4.4456 g AOT in isooctane (100 mL) to obtain a concentration of AOT isooctane solution at around 0.10 mol/L. Add dodecyl mercaptan (0.202 g) into an isooctane solution (100 mL) to form modifier solutions with concentrations of about 0.01 mol/L. We proceeded with the “reverse” diffusion with the isooctane solution of Au(III). The silver precursor (Ag(I)-OA) was obtained using phase transfer of 0.01 mol/L silver ion aqueous solution and 0.05 mol/L oleylamine ethanol solution. The concentration of sodium sulfide solution was 0.1 mol/L.

### 2.3. Preparation of Reversed-Phase Water Micelles

A single micelle system of water/AOT/isooctane was constructed using oil-soluble Au(III)-OA as the precursor and ascorbic acid as the reducing agent. An amount of 675 μL of a 0.60 mol/L solution of ascorbic acid was added to a 50 mL solution of AOT isooctane at room temperature and emulsified using a cell-shredding machine (ultrasonic time: 15 s, interval: 10 s, working: 10 times) to obtain reversed-phase water micelles.

### 2.4. Synthesis of Nanoparticles

An amount of 6000 μL of Au(III)-OA oil phase solution was added to the emulsion system and stirred for 2 h. Subsequently, 0.01 mol/L dodecyl mercaptan was introduced while stirring was continued for another 2 h. The resulting solution was initially subjected to distilled water treatment, followed by centrifugation at a speed of 10,000 rpm for 3 min to remove the organic matter at the interface between the two phases and lower aqueous phase. After being washed thrice with distilled water, it was further treated with ethanol until colorless. Finally, the precipitate was vacuum dried to obtain powdered products. Monodisperse AuNPs were obtained through one-time separation process with water and ethanol, and the steps of synthesis of Au-Ag alloy nanoparticles (Au-Ag NPs) and Ag_2_S quantum dots were the same as above.

### 2.5. Material Characterization

Transmission electron microscopy (TEM, JEOLJEM2100, JEOL, Tokyo, Japan) was used to characterize the morphology of the samples at an acceleration voltage of 100 kV. A small amount of prepared sample ethanol solution was taken using a pipette and dropped onto a carbon-film-coated copper grid three times, with each addition spaced 10 min apart. The sample was then observed after drying.

X-ray diffraction (XRD) was used to characterize the phase structures of the as-synthesized samples. The XRD pattern was recorded on a powder X-ray diffractometer (Ultima IV, Rigaku, Tokyo, Japan) by utilizing Cu Kα radiation, and scans were made from 30° to 90° (2θ) with a speed of 10 °C/min. The operating voltage was 36 kV, and the current was 30 mA.

UV–vis absorption spectra were recorded on a spectrophotometer (Youke UV-755B, Shanghai Youke Instrument Co., LTD, Shanghai, China). To record the position of the UV absorption peak of the nanoparticles, we used 3 mL of the synthesized solution and performed spectral analysis in the range of 360–800 nm using a 0.1 M AOT isooctane solution as the background.

Field emission scanning electron microscopy (SEM, Sigma500, Carl Zeiss AG, Jena, Germany) and energy dispersive analysis were used to characterize the surface appearance of the samples. The samples were characterized at an accelerating voltage of 15 kV. EDS provided information about the elemental composition of the samples.

X-ray photoelectron spectroscopy (XPS) analysis was performed using the Thermo Scientific Escalab 250Xi (Thermo-Fisher, Shanghai, China). The sample, after thorough drying, was affixed to a conductive adhesive and tested. XPS provided information about the elemental composition and chemical states of the elements present in the samples.

Fourier transform infrared spectroscopy (FTIR) were recorded using the Nicolet 670 Fourier (Thermo-Fisher, Waltham, MA, USA) transform infrared spectrophotometer. First, the background infrared spectrum was recorded by adding isooctane or ethanol to the sample chamber. After cleaning the sample chamber, the prepared sample of isooctane solution or ethanol solution was added, and the infrared spectrum of the sample was measured. The FTIR spectra provided information about the functional groups present in the samples.

Differential scanning calorimetric (DSC, DSC250, Waters, Milford, MA, USA) data were recorded by increasing the temperature from 30 °C to 550 °C at a heating rate of 10 °C/min under the nitrogen atmosphere. The data were recorded by a thermal analyzer (TA, SDT650, Waters, Milford, MA, USA) The electron coupled plasma parameters are as follows: the RF power was 1550 W, and the cooling gas flux was 15 L/min.

Zeta potential data were recorded with Dynamic Light Scattering (Zetasizer Nano ZSE, Malvern Panalytical Co., LTD, Malvern, UK).

Thermal infrared images were photographed with FLUKE Ti S20 infrared camera (Fluke Test Instrument Co., LTD, Shanghai, China).

### 2.6. Heating Effect Test

We established a comprehensive photothermal imaging system, which included a colorimetric dish, a laser illuminator, and a temperature-controlled chamber. In this system, the laser illuminator emitted near-infrared light (450 nm, 3 W) that uniformly irradiated the AuNPs (1 mg/mL) placed on the sample table. The resulting photothermal image was captured using specialized equipment for photothermal imaging. This integrated setup ensured stable positioning of both the laser illuminator and the sample.

Temperature rise experiment: The ambient temperature was maintained at 16.0 °C throughout the experiment conducted in a colorimetric dish, ensuring consistent conditions. A dispersion of 2 mg AuNPs in 2 mL deionized water was exposed to laser irradiation from an illuminator with an approximate irradiated area of 1 cm^2^. Following irradiation, temperature changes were recorded every 30 s using a Fluke infrared thermal imager.

Cyclic experiment: This experiment was performed within a cuvette. The temperature of the cupola was stabilized with 30 s irradiation first. After continuous illumination for 30 s using the lamp source, temperature changes were recorded, subsequently allowing it to cool down at room temperature for 90 s before recording further temperature changes. This cyclic process was repeated five times.

## 3. Results

### 3.1. The Structure of Gold Nanoparticles

The double-soluble Au(III)-OA with a water-to-oil phase-transfer rate of 88% in Table 1 was obtained by adding isooctane to the mixed solution of OA ethanol solution and Au(III) aqueous solution. The molar ratio of OA to Au(III) was 5:1 (Equation (1)). Size-controllable AuNPs were synthesized by tuning *w* in multiple mono-micelles prepared with VC and AOT. The transmission electron microscopy (TEM) images and corresponding particle size distribution histogram reveal that the AuNPs exhibit uniform particle size, controllable dimensions, and a self-assembled arrangement (Figure 1). The size of AuNPs prepared with the uni-micelle method is determined by the size of the reversed-phase micelles, which directly exhibit proportionality to *w* (Figure 1f). As *w* varies, the mean diameter of AuNPs is 4.56 nm, 5.97 nm, 7.62 nm, 9.53 nm, and 9.87 nm, respectively, with corresponding standard deviations (σ) of 9%, 8%, 7%, 8%, and 13% (Equation (2)), indicating a narrower distribution of AuNPs compared to previous reports [25,26,27]. The findings indicated that the size of AuNPs increased with an increase in *w*; however, the dispersity of the particles is worsened when *w* = 12.5 (Figure 1e). Based on the fitting of the first four data points, the relevant model (D (nm) = 0.71 *w* + 2.37) was established based on the experimental result for the uni-micelle method, which is consistent with Julian’s results [28] using aerosol reversed micelles under high pressure. In the second set of data, we replicated the experiment three times using *w* = 5 as an example. The results of all the experiments are similar (σ < 8%), which proves that monodisperse AuNPs can be synthesized with this method (Appendix A). Then, we replicated the experiments for all *w*, and the results of all the experiments are similar (σ = 10%, 7%, 10%, 7%, 10%), further confirming the relationship between particle size and water–oil ratio (Appendix A). The experimental results show that monodisperse AuNPs can be controlled synthesized with the size of 4–11 nm via this process; the adding rate and amount of Au (III) can be easily tuned; AuNPs can be obtained through a one-time separation process with a higher yield (61%), lowering the cost (Equation (3)).

X-ray diffraction (XRD), TEM, and selected area electron diffraction (SAED) were used to further reveal the crystal structure of the product. The chemical composition was characterized using ultraviolet–visible spectroscopy (UV), scanning transmission electron microscopy–energy dispersive X-ray spectroscopy (STEM-EDS), and X-ray photoelectron spectroscopy (XPS). The good crystalline properties of the AuNPs are shown in SAED (Appendix A), where the diffraction patterns proved the fcc structure of AuNPs, including the (111), (200), (220), and (311) planes. The lattice spacing of AuNPs (*w* = 5) is 0.235 nm, corresponding to the crystal face of Au (111), as can be seen in Figure 2a. The XRD pattern of AuNPs, shown in Figure 2b, exhibits five prominent peaks located at 38.22°, 44.48°, 64.46°, 77.46°, and 81.68°, assigned to the (111), (200), (220), (311), and (222) planes of a face-centered cubic (fcc) structure, respectively (JCPDS no. 99-0056) [29]. The clear lattice (Appendix A) is observed in larger particles (*w* = 10), indicating that AuNPs can maintain good crystallographic properties even with an increase in size. Moreover, large-area monodisperse AuNPs (Appendix A) can be synthesized using this method, while the structure of cyclic penta-twinned gold nanocrystals (Appendix A) can be observed [30]. The UV absorption peaks of the particles at different concentrations consistently appear around 524 nm [31,32], providing compelling evidence for the successful synthesis of small-size monodisperse AuNPs (Figure 2c). The solution transitioned from light pink to wine red with the increase in the concentration of Au(III), indicating the successful formation of AuNPs from a macroscopic perspective. This result is in agreement with that reported by Abbas [29]. This band gap value (Equation (4)) of 1.68 eV (Appendix A) is responsible for the better photothermal conversion efficiency of the synthesized AuNPs due to the easy separation of electron–hole pairs under sunlight. The full XPS spectrum of AuNPs reveals the presence of Au and S elements, while no N elements were detected, which confirms the removal of AOT (Figure 2d–f). The combination of Au 4f_5/2_ (88.0 eV) and Au 4f_7/2_ (84.2 eV) provides compelling evidence that the particles are Au (0). The electron interaction between Au and S induces shifts in the absorption peaks of S 2p_1/2_ (163.3 eV) and 2p_3/2_ (162.2 eV) as well as two smaller peaks at 86.2 eV and 82.7 eV, indicating the successful synthesis of thiolated AuNPs [33,34]. The STEM-EDS images (Appendix A) demonstrate a uniform distribution of Au and S, which means the successful synthesis of thiol-modified AuNPs [35]. In the Appendix A, AuNPs reduced by ascorbic acid exhibit better morphology and monodispersity. Additionally, their synthesis solution appears darker within the same reaction time. This can be attributed to the strong reducing ability of ascorbic acid, which contains hydroxyl (-OH) groups that are easily ionized. In redox reactions, ascorbic acid can lose two hydrogen ions, which possess strong reducing properties (Equation (5)). The controlled synthesis of Au-Ag NPs [15] (Appendix A) and silver sulfide (Ag_2_S) semiconductor quantum dots (Figure 2g–i) was preliminarily explored based on the experimental verification of the synthesis mechanism of size-controlled AuNPs. Differential scanning calorimetry (DSC) results (Appendix A) showed that AuNPs exhibit three peaks at 120 °C, 290 °C, and 460 °C, which are responding to the sintering temperature, removal temperature of mercaptan ligand, and melting temperature, respectively. The data are consistent with the previously reported data, which reported a melting temperature of about 465 °C, a desorption temperature of mercaptan molecules at about 200 °C, and sintering occurring near 150 °C [21]. The temperature shifts may be attributed to the following reasons. (a) Particle size: Smaller nanoparticles typically exhibit higher surface energy, leading to a lower sintering temperature and melting point. (b) Surface modification: The bonding strength of the particle with the ligand influences the rate of atomic diffusion during the sintering process, thereby affecting the sintering temperature.

### 3.2. The Mechanism of the Method

The phase-transfer effect of various amines on Au(III) and the concentration of the Au(III)-OA precursor solution were determined using an inductively coupled plasma optical emission spectrometer (ICP-OES). To achieve a high transfer rate of 88%, OA was found to be the optimal phase-transfer agent (Table 1). The reaction mechanism (Equation (5)) is shown in Figure 3. The alkyl group functions as an electron-donating moiety, augmenting the electronegativity of the nitrogen atom to generate RNH_3_^+^. When OA comes into contact with Au(III), the coordination of RNH_3_^+^ with Au(III) results in the formation of Au(III)-OA through a phase transfer [22,36]. When the equilibrium is reached in the diffusion of Au(III) between two phases, the concentration ratio of Au(III) in the oil phase to that in the aqueous phase is referred to as the diffusion equilibrium constant (K_1_ = 6.95). Subsequently, a reversed-phase transfer can be accomplished by combining the isooctane solutions of the complex with a fresh aqueous phase. The value of K_2_ (11.81) approximately equals K_1_, providing additional evidence for the bi-solubility exhibited by the Au(III)-OA (Table 2).

XPS and Fourier transform infrared (FTIR) spectroscopy were used to study the structure of the products at different reaction stages for a synthesis mechanism. The full XPS spectrum of Au(III)-OA (Appendix A) shows a pair of peaks at the positions of approximately 89.7 and 86.0 eV for Au 4f_5/2_ and Au 4f_7/2_ states, respectively, while a pair of peaks is located at approximately 400.9 and 399.4 eV for Au-N and C-N of N 1s states, respectively. The peaks of different chemical shifts are attributed to the electronic interaction between Au and N, which confirms the formation of Au(III)-OA [33,37]. By comparing the FTIR spectrum of OA and Au(III)-OA (Appendix A), we show that there are shifts at 3300 cm^−1^ (N-H), 1650 cm^−1^, and 1550 cm^−1^ (C=C), which can be attributed to alterations in electron cloud density resulting from the coordination of Au(III) with OA through the sharing of lone pair electrons on N [29]. The formation of Au(III)-OA can be explained by XPS and infrared spectroscopy. The AOT isooctane solution, containing VC, can effectively form stable water reversed micelles of a small size under the influence of an ultrasound. Furthermore, the hydrophilic end of AOT is attached to the surface of reversed-phase micelles. The interior of the reversed-phase micelle exhibits electropositivity as a result of Na^+^ ionization by AOT, while the water shell of the micelle displays electronegativity due to SO_3_H^−^. The size of the reversed micelle can be controlled by tuning w. The Au(III)-OA introduced into the system can be attracted by SO_3_H^−^ to form a more stable Au(III)-AOT complex. Due to its lipophilicity, OA is confined to the outer side of the reversed-phase micelle, enhancing the rigidity of the micelle to avoid accumulation. At this stage, Au(III) transfers from the oil phase to the water reversed-phase micelle and undergoes reduction via VC, leading to the formation of AuNPs, which is different from Wang’s previous work, where they used hexadecyltrimethylammonium bromide to achieve the complete transfer of AuNPs from the aqueous phase to the organic phase [38]. In contrast, our uni-micelle method is based on a reversible phase-transfer mechanism, and Au(III) is endowed with water–oil dual solubility through amine complexation. The experimental photo of Au(III)-AOT precipitation (Appendix A) shows the presence of precipitation in the mixed oil solution of Au(III)-OA and AOT, indicating that Au(III)-AOT exhibits greater stability compared to Au(III)-OA. The presence of 1300 cm^−1^ (O=S=O) in the FTIR spectra (Appendix A) confirms that AOT has been introduced into the system. The shifts at 3300 cm^−1^ (C-C) and 1750 cm^−1^ (C=O) relative to Au(III)-OA confirm the formation of Au(III)-AOT, while the absence of an absorption peak at 3300 cm^−1^ (N-H) can be attributed to the stronger nature of Au-S bonds (125 kJ/mol) compared to Au-N bonds (44 kJ/mol), which indicates that Au(III) dissociates from OA and associates with AOT [22,37]. The stable Au-S bonds are formed when thiol contacts Au in reversed-phase micelles. The FTIR spectra of AuNPs show that the coordination of Au and thiol (-SH) leads to shifts at 2900 cm^−1^ (C-H), 1300 cm^−1^, and 1000 cm^−1^ (C-S), which indicate the synthesis of thiol-modified AuNPs. Furthermore, AuNPs dispersed in water exhibit the potential of 56.2 mV (Appendix A), demonstrating similar electrical properties to those previously reported (25.7 mV, 36.8 mV) [31,32]. This may be attributed to the excessive amount of ascorbic acid aqueous solution being acidic, with the reduced AuNPs possibly adsorbing H^+^ on the surface during growth and the AuNPs adsorbing free Na^+^ from water micelles before modification [31]. When the liquid phase system is alkaline, the Zeta potential of AuNPs may show a negative value. In the next step, we will conduct a more detailed exploration of the effects of the solution environment [39].

### 3.3. Result of Heating Effect

As shown in Figure 4a, the temperature of pure water changed very little, while the aqueous solution of AuNPs exhibited a significant temperature rise from room temperature (16.0 °C) to 39.7 °C within just five minutes, surpassing previous reports on AuNPs (25 °C to 30 °C, 5 min) [40]. This improved performance can be attributed to the LSPR enabling the small-size particles to exhibit more efficient absorption. The procedure documented with infrared thermal imaging photographs (Figure 4a) shows that the temperature rise effect of AuNPs remained consistent throughout the cyclic experiment, providing evidence for the structural stability of AuNPs (Figure 4b,c). As depicted in Figure 4d, the TEM characterization of AuNPs recovered after five photothermal cycles experiments reveals that they still maintain a good microstructure, demonstrating the thermal stability of AuNPs and their ability for repeated use.

### 3.4. Formatting of Mathematical Components

The calculation of the ratio *w* of water to AOT is as follows:
(1)$$w=\frac{n_{H_{2}O}}{n_{AOT}}=\frac{\rho V_{H_{2}O}}{cV_{AOT}M_{H_{2}O}}$$where ρ is the density of water, $V_{H_{2}O}$ is the volume of water added, c is the concentration of AOT isooctane solution, $V_{AOT}$ is the volume of AOT isooctane solution added, and $M_{H_{2}O}$ is the molecular weight of water.The standard deviation (σ) is as follows:(2)$$\sigma ={\left[\sum {\left(D_{i}-D\right)}^{2}/\left(n-1\right)\right]}^{1/2}$$

The average particle size, D, and the corresponding particle size distribution, σ, were obtained from a few randomly chosen areas in the TEM image containing ~200 nanoparticles each. The size distribution is derived from histograms, which are obtained by measuring the diameter, D_i_, of all the particles from different parts of the grid to establish the histogram.

3.The yield of the obtained AuNPs is as follows:(3)$$\mathrm{Yield}=\frac{\mathrm{weight}\;\mathrm{of}\;\mathrm{AuNPs}}{{\mathrm{weight}\;\mathrm{of}\;\mathrm{Au}\;\mathrm{in}\;\mathrm{HAuCl}}_{4}\cdot 4H_{2}O}$$4.The optical band gap ($E_{g}$) of the synthesized AuNPs is as follows:(4)$${\left(\alpha h\upsilon \right)}^{1/2}=A\left(h\upsilon -E_{g}\right)$$where α is the absorption coefficient, h is Planck’s constant, υ is frequency, and A is a proportionality constant.5.The synthesis mechanism of AuNPs is as follows:
AuCl_4_^−^(aq) + RNH_3_^+^(org) → RNH_3_^+^AuCl_4_^−^*m*RNH_3_^+^AuCl_4_^−^ + *n*C_20_H_37_NaO_7_S → (C_20_H_37_O_7_S^−^)*_n_*(Au^3+^)*_m_*↓ + *m*RNH_3_^+^ + 4*m*Cl^−^ + *n*Na^+^C_6_H_8_O_6_ → C_6_H_6_O_6_ + 2*e*^−^ + 2H^+^Au^3+^ + 3*e*^−^ → Au*x*Au + *y*RSH → Au*_x_*(RSH)*_y_*(5)

## 4. Conclusions

In conclusion, we reported a uni-micelle approach for synthesizing small-size monodisperse AuNPs with fcc structures. The transfer rate of Au(III)-OA from water to oil was 88% through a phase transfer. The uni-micelle system was constructed using an aqueous solution of ascorbic acid, AOT, and isooctane. By tuning the size of the micelles, monodisperse AuNPs with a size of 4–11 nm were controlled synthesized. Monodisperse AuNPs were obtained through a one-time separation process with a higher yield of 61%. The enhanced rigidity of the micelles via the enrollment of OA effectively inhibited the agglomeration of the micelles. The AuNPs are demonstrated to have an fcc structure, while the initial tests indicate that the photothermal efficiency of the AuNPs is relatively stable. The method offers a promising way for the controlled synthesis of small-size alloy nanoparticles and semiconductor heterojunction quantum dots.

## Figures and Tables

**Figure 1 nanomaterials-14-00900-f001:**
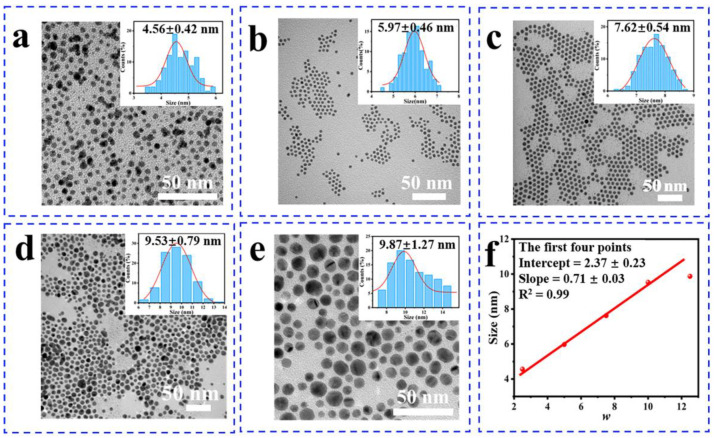
Morphological characterizations of AuNPs. (**a**–**e**) TEM and size histograms (*w* = 2.5, 5, 7.5, 10, 12.5). (**f**) The relevant model of the particle size and *w*.

**Figure 2 nanomaterials-14-00900-f002:**
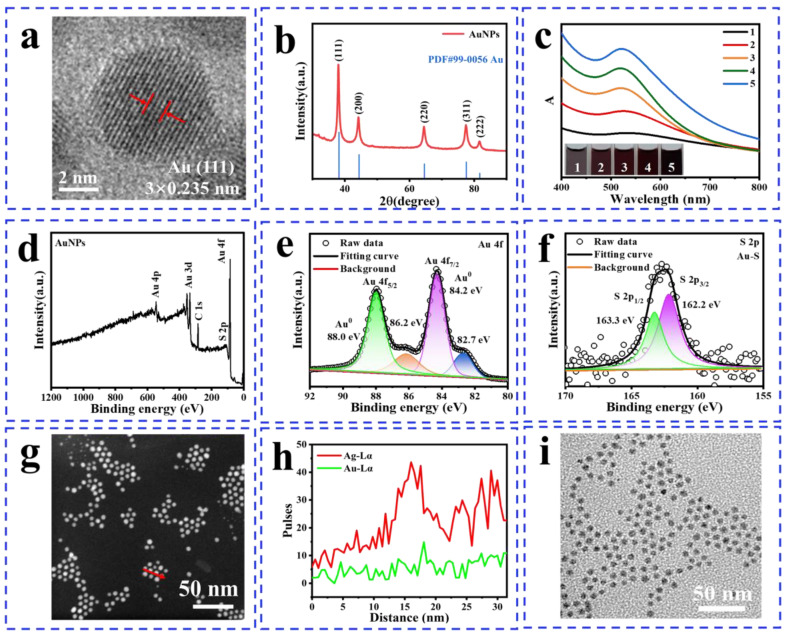
Structure and composition characterizations of nanoparticles (*w* = 5). (**a**) HRTEM image of AuNPs where the red arrows represent triple lattice spacing, (**b**) powder XRD pattern of AuNPs, (**c**) UV spectra, 1–5 represents Au(III)-OA transferred from HAuCl_4_ (10–50 mM), (**d**) the full XPS spectra of AuNPs, (**e**) XPS spectrum of Au, (**f**) XPS spectrum of S, the color green is used to represent high binding energy values, while pink is designated to signify low binding energy values, (**g**,**h**) TEM and EDS line scanning of Au-Ag NPs, and (**i**) TEM of Ag_2_SNPs.

**Figure 3 nanomaterials-14-00900-f003:**
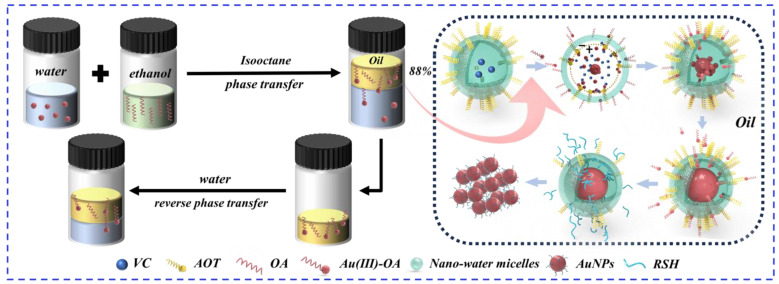
The proposed mechanism for the reversible phase-transfer, mechanism-mediated uni-micelle process.

**Figure 4 nanomaterials-14-00900-f004:**
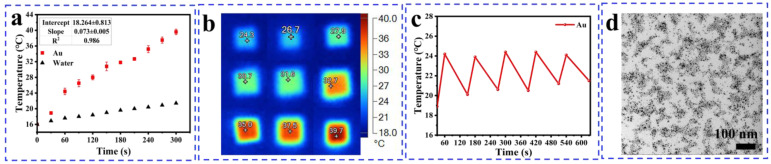
(**a**) Thermal infrared images of AuNPs (6 nm, 1 mg/mL) heated for 5 min under 450 nm laser irradiation at the power of 3.0 W, (**b**) heating curves of AuNPs, (**c**) temperature variations of AuNPs for five cycles, (**d**) TEM images of AuNPs recovered after five cycles of recycling experiments.

**Table 1 nanomaterials-14-00900-t001:** The rates of the water-to-oil phase transfer of gold ions in various amines. Phase-transfer rate is the ratio of the concentration of Au in oil phase after a phase transfer to the concentration of Au in the water phase after the phase transfer. The molar ratio of amidogen to Au(III) is 5:1.

Transfer Rate	Hexylamin	Octylamine	Decylamine	Dodecylamine	Oleylamine
Au(III)	36%	46%	64%	60%	88%

**Table 2 nanomaterials-14-00900-t002:** The data for the reversible phase transfer of Au(III)-OA.

Ion Specie	C_0_ (mM)	C_oil_ (mM)	C_aq_ (mM)	K_1_	C’_oil_ (mM)	C’_aq_ (mM)	K_2_
Au(III)	10.0	7.94	1.14	6.95	6.67	0.96	6.93

The metal ions come from HAuCl_4_ (aq). C_0_ represents the metal ion concentration of the original aqueous solution. C_oil_ and C_ag_ represent the metal ion concentration in the upper isooctane phase and lower aqueous phase after the phase transfer. K_1_ = C_oil_/C_ag_, a diffusion equilibrium constant. C’_oil_ and C’_ag_ represent the metal ion concentration in the upper isooctane phase and lower aqueous phase after the reversed-phase transfer. K_2_ = C’_oil_/C’_ag_.

## Data Availability

The data that support the findings of this study are available from the corresponding author upon reasonable request.

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
