# Peer review of "A Uni-Micelle Approach for the Controlled Synthesis of Monodisperse Gold Nanocrystals"

_nanomaterials, 2024, doi:10.3390/nano14110900_

Round 1
Reviewer 1 Report
Comments and Suggestions for Authors
The research paper presents a pioneering method, the uni-micelle approach, for the precise synthesis of monodisperse gold nanocrystals. This innovative technique leverages a single micelle housing a reductant in an aqueous solution and an oil-soluble Au(III)-OA complex, facilitating the creation of size-controlled AuNPs with remarkable monodispersity. The significance of this study lies in its potential applications across diverse fields, underscoring the necessity for cost-effective and efficient synthesis routes. Employing comprehensive experimental methodologies, including TEM imaging and photothermal imaging, the paper effectively validates the successful synthesis and characterization of monodisperse gold nanocrystals.
List of Suggestions for Improvement:
1. Scalability Concerns: While the paper admirably outlines the uni-micelle method's efficacy in laboratory settings, it lacks extensive discussion regarding its scalability for industrial applications. Addressing this limitation would bolster the practical utility of the proposed technique and enhance its attractiveness for commercial adoption.
2. Optimization Opportunities: Further exploration into optimizing the uni-micelle approach is warranted to augment both yield and reproducibility in the synthesis process. By fine-tuning experimental parameters and elucidating the underlying mechanisms governing nanoparticle formation, researchers can refine this method to achieve heightened efficiency and consistency.
3. Stability Assessment: The study would benefit from an investigation into the stability of the synthesized gold nanocrystals under varied environmental conditions. Assessing their robustness against factors such as pH fluctuations, temperature variations, and exposure to different solvents would yield invaluable insights for real-world applications, guiding researchers towards tailored nanoparticle design for specific contexts.
4. Figure Review Request: It would be valuable to conduct an extensive review of the article's figures to enhance aesthetics, organization, visibility, and understanding for readers. Clear and well-organized figures are essential for effectively conveying the experimental procedures and results to the audience, thereby improving the overall readability and impact of the paper.
In summary, while "A uni-micelle approach for controlled synthesis of monodisperse gold nanocrystals" presents a groundbreaking methodology with profound implications, addressing the outlined limitations through further research and refinement would bolster its significance and applicability in both academic and industrial settings.
Comments on the Quality of English LanguageMinor editing of English language required
Author Response
Dear reviewer,
Thank you very much for taking the time to review this manuscript. Please find the detailed responses below and the corresponding revisions in the resubmitted files.

Reviewer 2 Report
Comments and Suggestions for Authors
The authors have synthesized monodispersed gold nanocrystals by uni-micelle method. The authors have demonstrated that the AuNPs size of 4-11 nm have been synthesized by tuning the size of the micelles. Moreover, the monodisperse AuNPs have been obtained through one-time separation process with higher yield of 61%. Overall, this work can inspire more synthetic ideas of alloy nanoparticles. Therefore, I would like to recommend this work to publish in Nanomaterials. Below are some comments for the authors.
1. For “2.1. Materials”, the authors have described “Dodecylthiol, oleylamine (OA), hexylamine, n-Octylamine, decylamine, Dodecylamine, isooctane, glucose, ascorbic acid and ethanol were all of analytical grade, silver tri-fluoroacetate (98%), hydrogen tetrachloroaurate(III) (99.0%), sodium sulfide (99.9%), sodium bis (2-ethylhexyl) sulfosuccinate (AOT, 98.0%).” Please revise this sentence with correct English grammar.
2. From “2.5. Transmission electron microscopy (TEM)” to “2.14. Infrared Thermography”, these sections can be combined in one section “Material Characterization”
3. Figures should appear after the main text mentioned the figure. For example, Figure 2 should be placed after paragraph mentioned Figure 2.
4. For the introduction “The Gold nanoparticles have received widespread attention in various fields…,” for catalysis, more references could be cited to broaden the introduction.
https://doi.org/10.1021/acsmaterialslett.2c00752
Author Response

(The authors gave the same response as above.)

Reviewer 3 Report
Comments and Suggestions for Authors
This manuscript describes the synthesis of size-controlled monodisperse AuNPs with the size of 5-11 nm using uni-micelle approach. The method offers a promising route for the controlled synthesis of small-size alloy nanoparticles and semiconductor heterojunction quantum dots. Albeit the topic is in general of great interest, the following issues should be considered before consideration for publication:
1. Abbreviations such as OA should be defined at first mention.
2. Figure 1 should be enlarged such that the insets become clearer.
3. Authors should strengthen their introduction and discussion by considering other relevant works on Au NPs (10.1039/C8CP05154B, 10.1021/ja039874m, 10.1039/C9CP00056A) and include references on the use of AuNPs for applications such as these works ( Gold nanoparticles decorated graphene as a high performance sensor for determination of trace hydrazine...), and (monodispersed gold nanoparticles decorated carbon nanotubes as an enhanced sensing platform for nanomolar detection of tramadol)
4. It would be interesting to calculate the optical band gap (Eg) of the synthesized Au NPs from the UV-Vis data using Scherrer’s equation as in this work (Sunlight-assisted green synthesis of gold nanocubes using horsetail leaf extract: A highly selective colorimetric sensor for Pb2+….). Also, you can refer to this work in the UV-Vis, XRD FTIR discussions.
5. It is useful to discuss the positive zeta potential and its interpretation with respect to the charge on the particles surface.
6. Would it be possible to provide DLS hydrodynamic size of these particles.
7. Typo errors e.g. L25 (Gold)
Comments on the Quality of English Languageminor editing
Author Response

(The authors gave the same response as above.)

Round 2
Reviewer 1 Report
Comments and Suggestions for Authors
The reviewers have conducted an exceptionally detailed and insightful review, which not only met but exceeded our expectations. Their comments and suggestions were highly constructive and demonstrated a deep understanding of the subject matter. The comprehensive nature of their feedback significantly enhanced the quality of the manuscript.
Given the excellence of the review and the fact that all suggestions and comments have been meticulously addressed, I am pleased to accept the article in its current form.
Congratulations to the authors for their outstanding work and for contributing valuable research to the field of nanomaterials.
Reviewer 2 Report
Comments and Suggestions for Authors
The comments raised from the reviewers have been addressed, The manuscript can be published as its current form in Nanomaterials.
Reviewer 3 Report
Comments and Suggestions for Authors
The authors have addressed all of my comments and improved their manuscript.